# Letrozole Accelerates Metabolic Remodeling through Activation of Glycolysis in Cardiomyocytes: A Role beyond Hormone Regulation

**DOI:** 10.3390/ijms23010547

**Published:** 2022-01-04

**Authors:** Jun H. Heo, Sang R. Lee, Seong Lae Jo, Hyun Yang, Hye Won Lee, Eui-Ju Hong

**Affiliations:** 1College of Veterinary Medicine, Chungnam National University, Daejeon 34134, Korea; heojh94@o.cnu.ac.kr (J.H.H.); srlee5@cnu.ac.kr (S.R.L.); jsr7093@o.cnu.ac.kr (S.L.J.); 2KM Convergence Research Division, Korea Institute of Oriental Medicine, Daejeon 34054, Korea; hyunyang@kiom.re.kr (H.Y.); anywon1975@gmail.com (H.W.L.)

**Keywords:** letrozole, β-oxidation, glycolysis, cardiac hypertrophy

## Abstract

Estrogen receptor-positive (ER+) breast cancer patients are recommended hormone therapy as a primary adjuvant treatment after surgery. Aromatase inhibitors (AIs) are widely administered to ER+ breast cancer patients as estrogen blockers; however, their safety remains controversial. The use of letrozole, an AI, has been reported to cause adverse cardiovascular effects. We aimed to elucidate the effects of letrozole on the cardiovascular system. Female rats exposed to letrozole for four weeks showed metabolic changes, i.e., decreased fatty acid oxidation, increased glycolysis, and hypertrophy in the left ventricle. Although lipid oxidation yields more ATP than carbohydrate metabolism, the latter predominates in the heart under pathological conditions. Reduced lipid metabolism is attributed to reduced β-oxidation due to low circulating estrogen levels. In letrozole-treated rats, glycolysis levels were found to be increased in the heart. Furthermore, the levels of glycolytic enzymes were increased (in a high glucose medium) and the glycolytic rate was increased in vitro (H9c2 cells); the same was not true in the case of estrogen treatment. Reduced lipid metabolism and increased glycolysis can lower energy supply to the heart, resulting in predisposition to heart failure. These data suggest that a letrozole-induced cardiac metabolic remodeling, i.e., a shift from β-oxidation to glycolysis, may induce cardiac structural remodeling.

## 1. Introduction

Estrogen receptor-positive (ER+) breast cancer is the most common molecular subtype—accounting for approximately 80% of all breast cancer cases—that predominantly affects premenopausal women [1,2]. Anti-estrogen therapy is recommended as the primary adjuvant treatment for breast cancer patients after surgery [3]. Aromatase inhibitors (AIs)—including anastrozole and letrozole—and selective estrogen receptor modulators (SERMs), such as tamoxifen, are the commonly used hormonal drugs [4]. Currently, AI-based therapy is preferred over tamoxifen-based therapy as AIs are associated with a lower tumor recurrence rate than SERMs [5,6,7]. It has been previously established that AIs have better therapeutic efficacy against ER+ breast cancer than SERMs.

As a first line of breast cancer therapy, chemotherapy drugs are toxins and can induce cardiac toxicity [8,9]. As a result of chemotherapy, heart function could reduce and prevent enough oxygen and nutrients from supplying the body [10]. Along with chemotherapy drugs, anti-estrogenic therapy could be selected in ER+ breast cancer patients. Aromatase inhibitors are helpful for patients who have gone through menopause because they stop most estrogen production in the body, including in ovaries and fat [11]. However, cohort studies have reported that patients taking AIs—rather than tamoxifen—are more predisposed to cardiovascular diseases, including myocardial infarction and heart failure [12,13,14]. The underlying cause has been attributed to changes in lipid metabolism in the heart in response to AI-induced reduction in circulating estrogen levels [15,16]. This perspective seems reasonable, as estrogens play a role in energy and metabolic homeostasis [17]. Thus, the reduced circulating levels of estrogens can induce lipid metabolic changes [18]. This altered lipid metabolism could adversely impact heart function, as the heart’s high energy demands are met through fatty acid (FA) oxidation, which accounts for 60–80% of ATP production [19,20]. If the heart is not supplied with ATP from FA oxidation, glucose consumption increases (a lower ATP-yielding process in comparison) to meet energy needs [21,22,23]. Subsequently, cardiac metabolism is remodeled to use glucose as the primary source of energy, thereby promoting functional heart failure—including compensatory hypertrophy—due to unmet energy needs [24,25,26,27].

In line with this background, our study showed hypertrophic changes in the left ventricle and metabolic energy changes associated with decreased FA oxidation and increased glycolysis in the heart of female rats exposed to the AI letrozole. Although both AIs and tamoxifen are involved in estrogen receptor signaling, a previous cohort study revealed that AIs seem to be associated with more aggressive adverse effects on the heart compared to tamoxifen. Eventually, we intend to reveal the mechanism by which letrozole adversely affects cardiac metabolism and whether this mechanism involves its function as a hormone regulator or as an independent molecule, and to elucidate the mechanism by which cardiac metabolic remodeling leads to cardiac structural remodeling.

## 2. Results

### 2.1. Letrozole Promotes Hypertrophic Cardiomyopathy of the Left Ventricle (LV) in Letrozole-Treated Rats (LTZ-Rats)

To investigate whether letrozole acts as a causative factor of cardiovascular diseases—as is statistically indicated—slow-release letrozole pellets were inserted subcutaneously in female rats (Figure 1A). After exposing the rats to 0.1 mg/kg/day of letrozole for 4 weeks, the principal female reproductive organs, i.e., the ovaries and the uterus, were assessed to check whether the circulating estrogen levels were reduced. The organs were expected to show features of polycystic ovarian syndrome in response to reduced estrogen levels [28,29,30]. As expected, the ovaries became more swollen and reddish, and the diameter of the uterus became shrunken, in LTZ-rats compared to those in vehicle-treated rats (Veh-rats) (Figure 1B). Microscopic observation showed that the swollen ovaries bore multiple cysts and the endometrial thickness was reduced in the shrunken uterus of LTZ-rats compared to Veh-rats (Figure 1B). A quantitative analysis indicated that the ovary weight to body weight ratio was increased in LTZ-rats compared to that in Veh-rats (Figure 1C). In contrast, uterus weight to body weight ratio decreased in LTZ-rats compared to that in Veh-rats (42%, *p* < 0.05), along with a significant decrease in endometrial thickness in LTZ-rats compared to Veh-rats (49%, *p* < 0.05) (Figure 1D,E). Subsequently, quantitative RT-PCR analysis was performed. In accordance with the role of letrozole as an AI, mRNA expression levels of estrogen-responsive genes—including *prostaglandin D2 synthase* (*Ptgds*), *small heterodimer partner* (*Shp*) and *signal transducer and activator of transcription 3* (*Stat3*)—within the uterine tissues remarkably decreased in LTZ-rats compared to those in Veh-rats (*Ptgds*, 4%, *p* < 0.05) (*Shp*, 20%, *p* < 0.05) (*Stat3*, 61%, *p* < 0.05) (Figure 1F). The reduced levels of circulating estrogens, as indicated by the above mentioned factors, resulted in an increase in the body weight of LTZ-rats compared to that of Veh-rats (1.30-fold, *p* < 0.05) (Figure 1G). However, despite the increased body weight, no hematological differences—blood glucose, triglyceride (TG), and total cholesterol (TCHO) levels—were observed between Veh-rats and LTZ-rats (Figure 1H).

Following validation of the LTZ-challenged animal model, we examined the pathophysiological condition of the heart. The nuclei of LV cardiomyocytes were flatter in LTZ-rats than in Veh-rats, and the cytoplasm-to-nucleus ratio occupied the same area in LTZ-rats, compared to Veh-rats (86%, *p* < 0.05) (Figure 2A). As the above histological characteristics pertain to hypertrophic cardiomyopathy [31], we examined the expression levels of heart failure markers atrial natriuretic peptide (ANP) and brain natriuretic peptide (BNP). Interestingly, the expression levels of both ANP and BNP were significantly higher in LTZ-rats than in Veh-rats (ANP, 4.92-fold, *p* < 0.05) (BNP, 2.26-fold, *p* < 0.05) (Figure 2B). Subsequently, we performed quantitative RT-PCR to analyze the mRNA expression levels of cardiac physio-functional genes, including *ATPase sarcoplasmic/endoplasmic reticulum Ca2+ transporting 2* (*Atp2a2*), *myosin heavy chain, α isoform* (*Myh6*), and *myosin heavy chain, β isoform* (*Myh7*). The mRNA expression level of *Atp2a2*, an indicator of cardiac contractility, increased in LTZ-rats compared to that in Veh-rats (1.88-fold, *p* < 0.05) (Figure 2C). Additionally, *Myh6* and *Myh7* expression, reflecting increased cardiomyocyte size and LV wall thickening, respectively, increased in LTZ-rats compared to Veh-rats (*Myh6*, 1.19-fold, *p* < 0.05) (*Myh7*, 1.28-fold, *p* < 0.05) (Figure 2C). We also analyzed the mRNA expression of these three genes in vitro using the H9c2 cell line and observed a similar increase in expression in letrozole-treated cells (LTZ-cells) compared to that in vehicle-treated cells (Veh-cells) (*Atp2a2*, 1.20-fold, *p* < 0.05) (*Myh6*, 1.19-fold, *p* < 0.05) (*Myh7*, 1.28-fold, *p* < 0.05) (Figure 2D). However, no significant change in mRNA expression levels was observed in cells treated with ethynylestradiol (E2-cells) and testosterone (T-cells) (Figure 2D). These histological and molecular alterations in cardiomyocytes indicate that letrozole may induce hypertrophic cardiomyopathy of the LV.

### 2.2. Letrozole Decreases FA Consumption in Cardiomyocytes While Regulating Estrogen Levels

Next, we looked at the reasons underlying the hypertrophic changes in letrozole-exposed hearts, considering that cardiac structural remodeling, including cardiac hypertrophy, might develop in response to altered energy metabolism [24,26]. We first looked for traces of altered lipid metabolism related to TG accumulation and FA uptake and oxidation, as FA serves as the primary energy source for the heart [19,20]. TG levels in the hearts were higher in LTZ-rats than in Veh-rats (2.4-fold, *p* < 0.05) (Figure 3A). Conversely, Western blotting revealed that the expression of cluster of differentiation 36 (CD36), a plasma membrane protein facilitating FA uptake in cells, significantly decreased in LTZ-rats compared to that in Veh-rats (41%, *p* < 0.05) (Figure 3B). Additionally, quantitative RT-PCR indicated that the expression of *adipose triglyceride lipase* (*A**tgl*), an enzyme that hydrolyzes TGs to diacylglycerols and serves as the rate-limiting step of lipolysis, decreased in LTZ-rats compared to that in Veh-rats (59%, *p* < 0.05) (Figure 3C). In line with the expression of *Atgl*, the expression of genes associated with β-oxidation, including *carnitine palmitoyl transferase I* (*Cpt1*), *very long-chain acyl-CoA dehydrogenase* (*Acadvl*) and *medium-chain acyl-CoA dehydrogenase* (*Acadm*) also decreased in LTZ-rats compared to that in Veh-rats (*Cpt1*, 75%, *p* < 0.05) (*Acadvl*, 81%, *p* < 0.05) (*Acadm*, 73%, *p* < 0.05) (Figure 3C).

To determine whether these changes in lipid metabolism were attributed to reduced estrogen levels or to letrozole, an in vitro assay—using LTZ-, E2-, and T-treated H9c2 cells—was performed. CD36 was expressed at higher levels in E2-cells than in Veh-cells (1.23-fold, *p* < 0.05) (Figure 3D). The transcript-level expression of *A**tgl* and β-oxidation genes, including *Cpt1*, *Acadvl* and *Acadm,* was also higher in E2-cells than in Veh-cells (*Cpt1*, 2.16-fold, *p* < 0.05) (*Acadvl*, 1.36-fold, *p* < 0.05) (*Acadm*, 1.58-fold, *p* < 0.05) (Figure 3E). However, no significant changes in protein and mRNA expression were observed in LTZ-cells compared to Veh-cells, even though *Cpt1* level increased in T-cells compared to Veh-cells (1.53-fold, *p* < 0.05) (Figure 3E). These findings indicate increased TG accumulation and decreased TG degradation and FA oxidation, thereby suggesting that FA consumption by cardiomyocytes is lower in the presence of letrozole-induced reduced estrogen levels.

### 2.3. Letrozole Induces Enhanced Glucose Consumption in Cardiomyocytes Independent of Its Role in Modulating Estrogen Levels

After looking at the changes in lipid metabolism, we examined glucose metabolism to determine whether glucose consumption increased to compensate for the decrease in FA consumption. The expression levels of hexokinase 2 (HK2)—which catalyzes the first step of glycolysis [32]—and pyruvate kinase muscle form 2 (PKM2)—which catalyzes the last step of glycolysis [33]—increased in LTZ-rats compared to those in Veh-rats (HK2, 1.82-fold, *p* < 0.05) (PKM2, 1.42-fold, *p* < 0.05) (Figure 4A). Additionally, the expression of pyruvate dehydrogenase (PDH)—the enzyme that converts pyruvate to acetyl-CoA [34,35]—also increased in LTZ-rats compared to Veh-rats (2.03-fold, *p* < 0.05) (Figure 4A). From an in vitro assay using the H9c2 cell line, the expression levels of HK2, PKM2, and PDH were found to be higher in LTZ-cells compared to those in Veh-cells (HK2, 2.00-fold, *p* < 0.05) (PKM2, 1.24-fold, *p* < 0.05) (PDH, 1.86-fold, *p* < 0.05) (Figure 4B). Although the expression of PKM2 and/or PDH also increased in E2-cells and T-cells compared to that in Veh-cells (E2-cells, PKM2, 1.16-fold, *p* < 0.05) (E2-cells, PDH, 1.35-fold, *p* < 0.05) (T-cells, PDH, 1.55-fold, *p* < 0.05), the extent of the increase was more notable in LTZ-cells than in E2-cells and T-cells (Figure 4B).

To determine whether letrozole induces increased glucose consumption in cardiomyocytes independent of its role in reducing estrogen levels, we performed an in vitro assay using H9c2 cells, wherein the cells were exposed to high and low concentrations of glucose. In the presence of high glucose concentrations (HG), the expression levels of HK2, PKM2, and PDH increased in letrozole-treated cells (HG-LTZ-cells) compared to Veh-cells (HG-Veh-cells) (HK2, 1.73-fold, *p* < 0.05) (PKM2, 1.30-fold, *p* < 0.05) (PDH, 1.61-fold, *p* < 0.05) (Figure 5A). When exposed to low glucose concentrations (LG), the expression of HK2 alone increased in LTZ-cells (LG-LTZ-cells) compared to that in Veh-cells (LG-Veh-cells) (2.19-fold *p* < 0.05) (Figure 5A). Subsequently, we performed glycolysis stress tests for quantitatively monitoring the glycolysis rate as an indicator of glucose consumption. In LTZ-cells, the extracellular acidification rate (ECAR) substantially increased compared to that in Veh-cells (2.17-fold *p* < 0.05) (Figure 5B). These findings indicate that letrozole enhances glycolysis in cardiomyocytes, independent of its role in modulating estrogen levels, to compensate for decreased FA oxidation.

## 3. Discussion

Previous cohort studies on breast cancer indicate that AIs increase the risk of heart failure by up to 86% and cardiovascular mortality up to 50%, compared with SERMs such as tamoxifen [12,13,14]. The underlying cause has been linked to lipid metabolism changes in the heart, induced by decreasing levels of circulating estrogens [15,16]. However, not many research groups have focused on the direct influence of letrozole in cardiac metabolism. After exposure to letrozole for four weeks, the rats’ hearts showed metabolic energy changes involving decreased FA oxidation and increased glycolysis. Taking into account the fact that AIs induce more adverse hypertrophic changes in the heart compared to tamoxifen, letrozole may function to mediate such changes, along with modulating the estrogen levels.

To confirm the reduced estrogen levels in the animal model post letrozole exposure, we assessed the female reproductive organs, i.e., the ovaries and the uterus. When the subcutaneously administered pellets of letrozole were released into the body, the reduced levels of circulating estrogens induced in turn were reflected in the form of atrophy of the uterus, as letrozole exposure represents hypoestrogenism and hyperandrogenism in females [36,37]. LTZ-rats showed multi-ovarian cysts and a light uterus with a thinning endometrium, similar to the findings in our previous studies [28,29,30]. Furthermore, we also observed decreased expression of estrogen-responsive genes in the uterus, along with increased body weight. Although no significant changes in hematological profiles, including blood glucose, TG and TCHO, were observed in LTZ-rats, this might be considered to be due to a shortened letrozole exposure duration, unlike our previous studies. Nevertheless, these findings indicate that estrogen levels were sufficiently reduced in the animal model.

Next, we examined whether letrozole induced cardiac structural remodeling. It is interesting to note that cardiomyocytes within the LV had a higher cytoplasm-to-nucleus ratio and flatter nuclei in LTZ-rats compared to those in the Veh-rats, an observation that indicates structural remodeling [31]. Moreover, this histological observation was supported by increased expression levels of heart failure markers [38], such as ANP and BNP, and physio-functional markers, such as *Atp2a2*, *Myh6*, and *Myh7*, which indicate contractility [39], cardiomyocyte size [40] and LV wall thickness [41], respectively. In parallel with the animal study, H9c2 cardiomyocytes treated with letrozole also showed increased expression of the above three physiological markers. However, the two sex steroid hormones, ethynylestradiol (E2) and testosterone (T), failed to change these physiological markers. These results suggest that letrozole could directly make the heart vulnerable to cardiac hypertrophy.

It has been reported that cardiac structural remodeling is mediated by a shift in cardiac energy metabolism, where glucose is utilized as the primary energy source instead of FAs [24,25,26,27]. LTZ-rats showed cardiac hypertrophy and TG accumulation, and decreased β-oxidation, TG degradation, and intracellular uptake of FA. As the estrogen receptor acts as a transcription factor for the *peroxisome proliferator-activated*
*receptor* (*PPAR*) *α* promoter [42], the expression of PPARα-regulated β-oxidative transcripts including *Cpt1*, *Acadvl* and *Acadm* decreased in the hearts of LTZ-rats where the circulating estrogen levels were low [43,44]. The expression of *A**tgl*—associated with TG degradation—was also decreased due to reduced estrogen receptor signaling [45], and reduced *A**tgl* led to excess TG accumulation in the heart [46]. Furthermore, TG accumulation prevents the intracellular uptake of FA by CD36 in cardiomyocytes [47]. These series changes in lipid metabolism were also observed in E2-treated H9c2 cells, but not in LTZ-treated cells. These findings suggest that estrogens can modulate lipid metabolism by regulating estrogen receptor signaling and PPARα signaling, and thereby, might induce cardiac structural remodeling via altered lipid metabolism. Therefore, it also might be intimated that PPARα agonists, including fenofibrate as an activator of fatty acid oxidation, might have an effect as a target drug for ER+ breast cancer, although this needs to be justified through further study [48,49].

In the presence of decreased FA oxidation, the heart increases glucose consumption to meet the energy demands during pathological conditions [50,51]. LTZ-rats showing decreased FA oxidation displayed increased glycolysis. As a causative factor of this increase, estrogens themselves could be excluded because estrogens are known to stimulate glycolysis [52]. It is expected for the cause to be a compensatory mechanism from estrogen-induced decreases in FA oxidation, and/or to be independent of its role in modulating estrogen levels. Interestingly, this induced glucose consumption was more notable in H9c2 cells treated with letrozole than in E2-treated cells. This suggests that letrozole increased glycolysis, independent of its role in modulating estrogen levels.

To determine whether letrozole enhances glucose consumption in cardiomyocytes, we performed a glucose addition assay and glycolysis stress test using H9c2 cells. The LTZ-cells showed increased glycolysis in the presence of high glucose concentration and increased ECAR, which can directly evaluate the amount of glucose consumed by the cells quantitatively [53]. This demonstrates that letrozole facilitates glucose consumption in cardiomyocytes independently in response to an estrogens-induced decrease in FA oxidation. This finding indicates that letrozole increases glycolysis to compensate for decreased FA oxidation and independently accelerates glucose consumption.

Our study demonstrated that letrozole, an AI, modulates cardiac metabolic remodeling in two ways. One is via estrogen-mediated lipid metabolism, and the other is via estrogen-independent glycolysis. Reduced estrogen levels induced by letrozole suppress fatty acid consumption, and the insufficient energy is compensated for through enhanced glucose consumption in cardiomyocytes. Additionally, letrozole directly induces glucose consumption, while estrogens itself do not mediate it. In such cardiac metabolic remodeling conditions, the heart is unable to meet its high energy demands, and therefore, is predisposed to cardiac structural remodeling, including concentric hypertrophy. To prevent cardiac remodeling, we suggest that ER+ breast cancer patients receiving letrozole should abstain from a carbohydrate-rich diet and medicate with an activator of fatty acid oxidation for facilitating fatty acid oxidation, at least during the letrozole treatment.

## 4. Materials and Methods

### 4.1. Antibody

Antibodies used in the present study are shown in Table 1 below.

### 4.2. Animals

Norway female rats were accommodated for one week and tested in the pathogen-free facility at Chungnam National University, in line with the Chungnam National University Animal Care Committee (202006A-CNU-104). We provided them with standard chow diet (LabDiet^®^ Rodent Diet 5001 contains as a percentage of total calories: 28% protein, 60% carbohydrate and 12% fat) and water ad libitum up to the age of eight weeks. At the age of four weeks, letrozole slow releasing pellets (0.1 mg/kg/day) were inserted subcutaneously in the nape. The standard chow diet was provided consistently for four weeks and the rats were then sacrificed. The number of rats used for the experiment was three and three for each group, designated Vehicle-rats (Veh-rats) and Letrozole-rats (LTZ-rats).

### 4.3. Hematological Measure Assay

Blood glucose was measured using Accu-Chek Active [Model GB] (CR2032, Roche Diabetes Care GmbH, Mannheim, Germany). Triglyceride (TG) was evaluated with non-diluted serum. Serum TG level was measured with FUJI DRI-CHEM SLIDE (TG-P III) by DRI-CHEM4000 (Fuji Film, Tokyo, Japan). Identically, total cholesterol (TCHO) level was measured FUJI DRI-CHEM SLIDE (TCHO-P III), after serum was diluted by 1/5 fold with sterilized phosphate-buffered saline (PBS).

### 4.4. Hematoxylin & Eosin Staining

Rats’ hearts were arranged in 10% formaldehyde solution and embedded in paraffin. The paraffinized samples were sectioned to 4 μm and stuck on slide glass. The samples were stained with haematoxylin and eosin (H&E) after being de-waxed and rehydrated. The stained samples were examined using a VM600 Digital Slide Scanning System (Motic, San Francisco, CA, USA).

### 4.5. Tissue Triglyceride Measure Assay

LV tissues of the hearts were homogenized with PBS and centrifuged at 13,000 rpm for 10 min. Supernatants were discarded and pellets were dried. After drying, the pellets were re-suspended in 200 μL PBS. An amount of 50 μL of resuspension samples was dispensed in a 96 well plate and 150 μL of enzyme reagent (AM157S, ASAN PHARM. CO., LTD., Seoul, Korea) was added. The plate was incubated at 37 °C for 1 h. After 1 h, relative absorbance value of the incubated samples was measured at 550 nm using a microplate reader (Infinite 200 PRO, Tecan Life Sciences, Männedorf, Switzerland).

### 4.6. Cell Culture

To validate in vivo experimental data, we performed in vitro assays using the H9c2 cardiomyocytes line. The cells were grown at 37 °C in a 5% CO_2_ atmosphere and were cultured with Dulbecco’s modified Eagle’s medium (DMEM)-High glucose (LM 001-05, Welgene, Gyeongsan, Gyeongbuk, Korea) added to 5% fatal bovine serum (FBS) and 1% penicillin-streptomycin (P/S). Additionally, to remove endogenous steroid hormones, a starvation medium was used, composed of DMEM/F-12 (LM 002-05, Welgene) added to 2% charcoal dextran fetal bovine serum (CD-FBS) and 1% P/S.

To assess fluctuations in lipid metabolism, we supplied fatty acids to cells. Fatty acids (Palmitic acid 330 μM, Oleic acid 660 μM) dissolved in absolute ethanol (K46253383 506, EMD Millipore Corp.) were treated and incubated for 48 h. Letrozole, ethynylestradiol and testosterone (Letrozole, L0248, Tokyo chemical industry CO. LTD., Tokyo, Japan) (Ethynylestradiol, E0037, Tokyo chemical industry CO. LTD) (Testosterone, T0027, Tokyo chemical industry CO. LTD) were dissolved in dimethyl sulfoxide (DMSO) (#MKCH9998, Sigma-Aldrich, St. Louis, MO, USA). The three reagents were added to 100 nM/L respectively and incubated for 24 h.

### 4.7. Glucose Addition Assay

After H9c2 cells were cultured with DMEM-High glucose (LM 001-05, Welgene) added to 5% FBS and 1% P/S, the medium was changed to a low glucose medium (LM 001-17, Welgene), which contained glucose 1000 mg/L, and was added to 2% CD-FBS and 1% P/S, similar to the above starvation medium. Following additional supplementation with fatty acids (Palmitic acid 330 μM, Oleic acid 660 μM), the cells were incubated for 48 h. At 24 h of incubation with the low glucose medium, extra glucose was supplemented to high glucose group cells using 45% D-glucose (LS 001-02, Welgene) to reach glucose concentrations of 4500 mg/L, and incubation was maintained for the remaining 24 h.

### 4.8. Glycolysis Stress Test

After preparing H9c2 cells incubated in a starvation medium to which fatty acids were added for 48 h and treated with letrozole for an additional 24 h, the cells were further incubated with a low glucose medium (LM 001-17, Welgene) free from FBS and P/S for 1 h. The medium was changed to XFp running buffer (1035757-100, Agilent Technologies, Santa Clara, CA, USA), which contains the identical amounts of D-glucose, sodium pyruvate and L-glutamine as a low glucose medium, to decarboxylate the cells for 1 h. Glucose (25 mM) was added and extracellular acidification rate (ECAR) was measured using a Seahorse XFp glycolytic rate assay kit (103022-100, Agilent Technologies) and Seahorse XFp analyzer (S7802A, Agilent Technologies).

### 4.9. Total RNA Extraction and Real-Time Quantitative PCR

Total RNA samples, including the LV tissues of the hearts and H9c2 cells, were extracted using TRIzol reagent (15596-026, Life Technologies, Carlsbad, CA, USA). cDNA was synthesized with 1.5 µg of total RNA samples and reverse transcriptase kit (SG-cDNAS100, Smartgene, Lausanne, Switzerland). Quantitative PCR (real-time PCR) was performed using Excel Taq Q-PCR Master Mix (SG-SYBR-500, Smartgene) and Stratagene Mx3000P (Agilent Technologies). All primers used for real-time PCR were produced by Bionics Inc. (Seoul, Korea) and are listed in Table 2. mRPLP0 was used as control for in vivo and in vitro samples. All experiments were run more than three times, and mRNA values were calculated based on the cycle threshold and monitored for an amplification curve.

### 4.10. Western Blotting

Protein samples, including LV tissues of the hearts and H9c2 cells, were extracted using T-PER reagent (78510, Thermo Fisher Scientific, Waltham, MA, USA), which is a protein lysis buffer. Through a Bradford assay, the sample concentration was quantified using PRO-Measure solution (#21011, Intron, Seoul, Korea). SDS-PAGE electrophoresis was performed on 10% polyacrylamide gels and the gels were transferred to a PVDF membrane. Following the transfer, the membranes were blocked using 3% BSA (9048-46-8, LPS Solution, Chisinau, Moldova) with diluted TBS-T buffer (04870517TBST4021, LPS solution). Primary antibodies (Table 1) dissolved with 3% BSA were operated overnight in 4 °C. The membranes were washed three times with TBS-T and secondary antibodies (Table 1) were operated in identical fashion to primary antibodies. The blotting results were detected with ECL solution (XLS025-0000, Cyanagen, Bologna, Italy) and Chemi Doc (Fusion Solo, Vilber Lourmat, Marne-la-Vallée, France).

### 4.11. Statistical Analysis

Data are reported as the mean ± standard deviation (SD). Using Graph Pad Software (Graph Pad Inc., San Diego, CA, USA), Student’s *t*-test and one-way ANOVA were performed to recognize differences between means.

## 5. Conclusions

Our data indicate that letrozole is an independent molecule that can promote glucose oxidation as well as an estrogen blocker in cardiomyocytes. As the heart typically uses fatty acid as a primary energy source, increased glycolysis is a risk factor for heart health. Interestingly, letrozole increased glycolysis in cardiomyocytes, along with cardiac hypertrophy. We found out that the increase in glucose consumption resulted from two motions: one was a compensatory reaction involving estrogen-mediated reduced fatty oxidation and the other was hormone-independent increased glycolysis. This study proposes perspectives on why ER+ breast cancer patients receiving letrozole are vulnerable to cardiovascular disease and how to establish precautions against cardiac adverse effects from this drug.

## Figures and Tables

**Figure 1 ijms-23-00547-f001:**
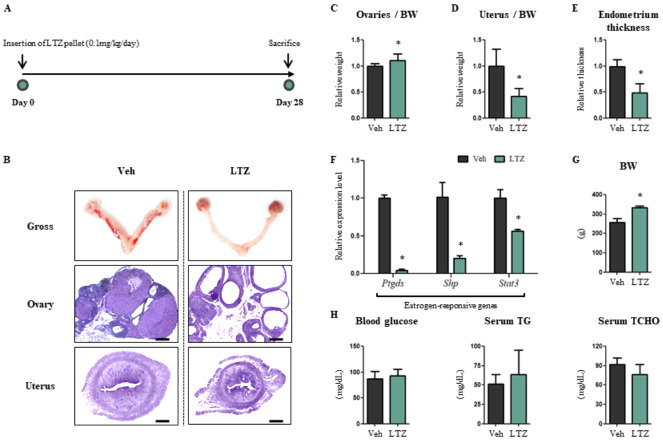
Letrozole creates a low estrogenic environment in female rats. (**A**) Schematic diagram shows the schedule of an animal experiment. Norway rats aged 3 weeks were acclimatized for a week and slow-releasing pellets which were either vehicle (Veh) or letrozole (LTZ) were inserted into subcutaneous parts for four weeks. (**B**) To validate fluctuation in the estrogenic environment, female reproductive tissues including ovaries and uterus were observed as gross and were examined via haematoxylin and eosin stain (H&E). Scale bar = 1000 µm. (**C**) Relative weight of ovaries to body weight was measured. (**D**) Relative weight of uterus to body weight was measured. (**E**) Endometrium thickness was measured quantitatively using Image J software. (**F**) mRNA levels of estrogen-responsive genes were determined via quantitative RT-PCR with uterus tissues. Mouse 60S acidic ribosomal protein P0 (mRPLP0) was used as internal control. (**G**) Body weight was measured at four weeks after pellets were inserted. (**H**) Hematological profiles including blood glucose, serum triglyceride (TG) and serum total cholesterol (TCHO) were measured. The values stand for mean ± S.D. * *p* < 0.05 compared to Veh-rats.

**Figure 2 ijms-23-00547-f002:**
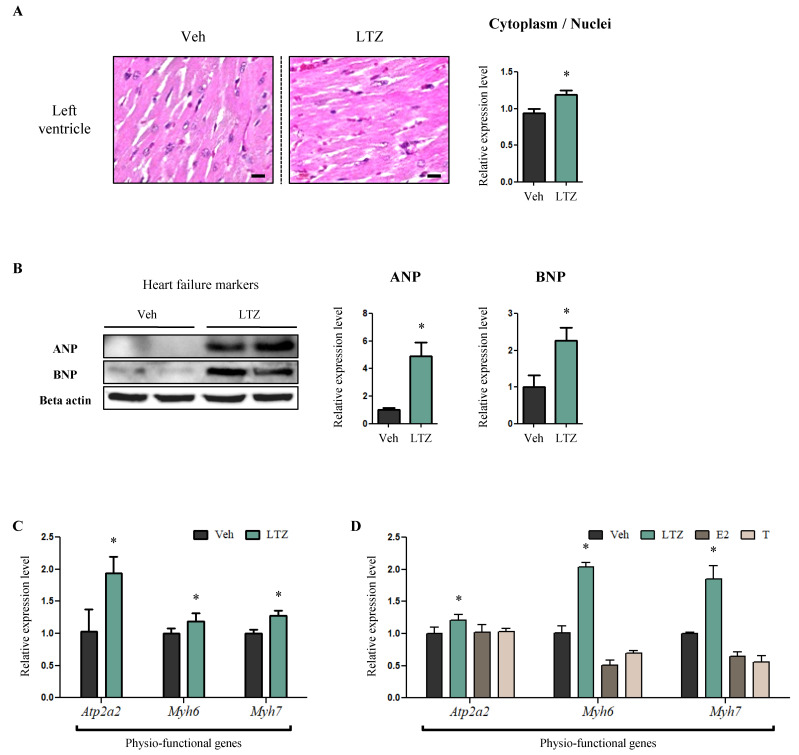
Letrozole induces hypertrophic change in the left ventricle (LV) in female rats. (**A**) The hearts of rats were examined via haematoxylin and eosin stain (H&E) to examine whether cardiac structural remodeling was developed when exposed to LTZ. Scale bar = 30 µm. Area of cytoplasm and nuclei were measured quantitatively using Image J software. (**B**) The protein levels of representative heart failure markers were determined by Western blotting with LV tissues. Beta actin was used as internal control. (**C**) mRNA levels of representative physio-functional genes were determined using quantitative RT-PCR, with LV tissues. mRPLP0 was used as internal control. (**D**) mRNA levels of representative physio-functional genes were determined using quantitative RT-PCR, with H9c2 cells. These cells were exposed to fatty acids (Palmitic acid 330 μM, Oleic acid 660 μM) and treated with LTZ, ethynylestradiol [E2] and testosterone [T], respectively. mRPLP0 was used as internal control. The values stand for means ± S.D. * *p* < 0.05 compared to Veh-rats or Veh-cells.

**Figure 3 ijms-23-00547-f003:**
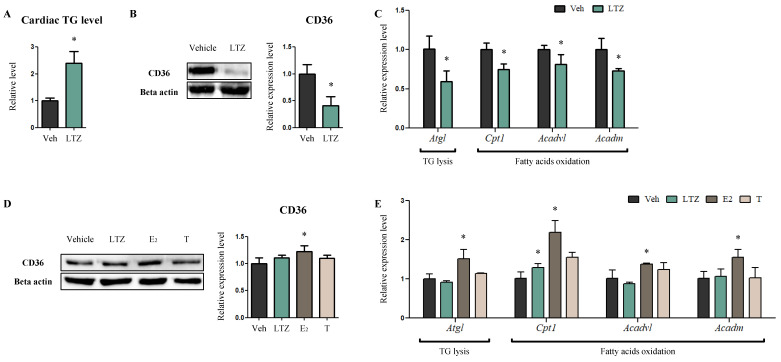
Letrozole alters all-around lipid metabolism, including TG accumulation and degradation and fatty acids uptake and oxidation in cardiomyocytes. (**A**) The degree of TG accumulation within LV was evaluated using a triglyceride measure assay kit. (**B**) The protein levels of fatty acids translocase were determined via Western blotting, with LV tissues. Beta actin was used as internal control. (**C**) mRNA levels of related genes of TG degradation and the β-oxidation gene were determined via quantitative RT-PCR with LV tissues. mRPLP0 was used as internal control. (**D**) The protein levels of fatty acids translocase were determined via Western blotting, with H9c2 cells. These cells were exposed to fatty acids (Palmitic acid 330 μM, Oleic acid 660 μM) and treated with LTZ, ethynylestradiol [E2] and testosterone [T], respectively. Beta actin was used as internal control. (**E**) mRNA levels of related genes of TG degradation and the β-oxidation gene were determined via quantitative RT-PCR with H9c2 cells, which was handled in the same fashion as (**D**). mRPLP0 was used as internal control. The values stand for means ± S.D. * *p* < 0.05 compared to Veh-rats or Veh-cells.

**Figure 4 ijms-23-00547-f004:**
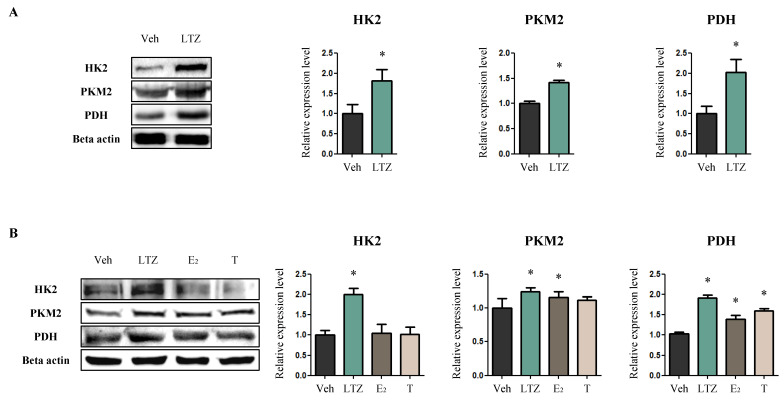
Letrozole increases glycolysis in cardiomyocytes. (**A**) The levels of proteins related to glycolysis were determined via Western blotting with LV tissues. Beta actin was used as internal control. (**B**) The levels of proteins related to glycolysis were determined via Western blotting with H9c2 cells. These cells were exposed to fatty acids (Palmitic acid 330 μM, Oleic acid 660 μM) and treated with LTZ, ethynylestradiol [E2] and testosterone [T], respectively. Beta actin was used as internal control. The values stand for means ± S.D. * *p* < 0.05 compared to Veh-rats or Veh-cells.

**Figure 5 ijms-23-00547-f005:**
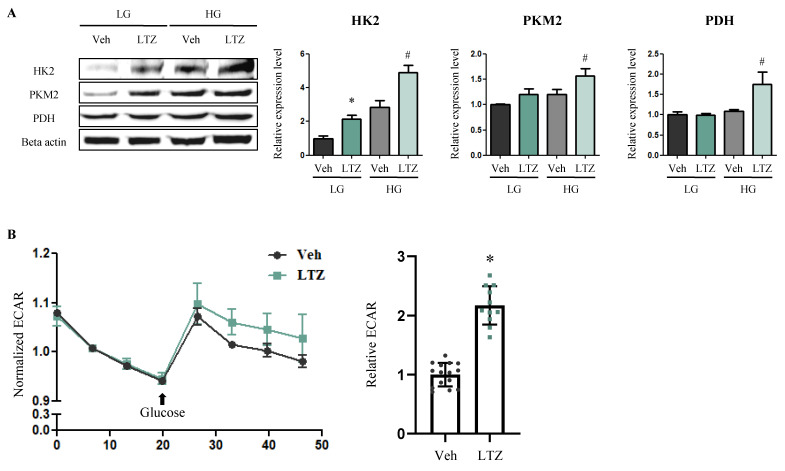
Letrozole directly increases the amount of glucose consumption in cardiomyocytes. (**A**) To evaluate whether glucose is consumed independent of letrozole role in modulating estrogens levels, we performed a glucose-addition assay with H9c2 cells treated with letrozole. Low glucose concentration (LG) and high glucose concentration (HG) contained 1000 mg/L and 4500 mg/L from D-glucose respectively. The levels of proteins related to glycolysis were determined via Western blotting. Beta actin was used as internal control. (**B**) To prove whether letrozole directly promotes glucose consumption, we performed a glycolysis stress test with H9c2 cells treated with letrozole. Extracellular acidification rate (ECAR) was measured quantitatively using a Seahorse XFp analyzer. The values stand for means ± S.D. * *p* < 0.05 compared to LG-Veh-cells or Veh-cells. # *p* < 0.05 compared to HG-Veh-cells.

**Table 1 ijms-23-00547-t001:** Antibodies used for experiment.

Primary Antibodies	Type	Cat.	Inc.
Beta actin	Mouse monoclonal	sc-47778	Santa Cruz biotechology
ANP	Rabbit polyclonal	ab14348	Abcam PLC
BNP	Rabbit polyclonal	ab19645	Abcam PLC
CD36	Rabbit polyclonal	A5792	Company ABclonal, Inc.
HK2	Rabbit polyclonal	A0994	Company ABclonal, Inc.
PKM2	Rabbit monoclonal	#4053T	Cell signaling technology
PDH	Rabbit polyclonal	A17432	Company ABclonal, Inc.
**Secondary antibody**	**Type**	**Code.**	**Inc.**
Anti-Mouse IgG	Goat	115-035-174	Jackonimmuno
Anti-Rabbit IgG	Mouse	211-032-171	Jackonimmuno

**Table 2 ijms-23-00547-t002:** Primers used for real-time PCR.

Gene Name	Upper Primer (5′-3′)	Lower Primer (5′-3′)	Species
*Rplp0*	AAA GGG TCC TGG CTT TGT CT	CCG ACT CTT CCT TTG CTT CG	Rat
*Ptgds*	GGT TCC GGG AGA AGA AAG AG	CAC TGA GAG GGA GTG GAA GC	Rat
*Shp*	CCT TGG ATG TCC TAG GCA AG	CAC CAC TGT TGG GTT CCT CT	Rat
*Stat3*	ATG AAG AGT GCC TTC GTG GT	TGT TCG TGC CCA GAA TGT TA	Rat
*Atp2a2*	CTG TGG AAA CCC TTG GTT GT	CTC CAA TGG GTG CAT AGG TT	Rat
*Myh6*	ATG ACC TTC AGG CTG AGG AA	CTC TCC TGG GTC AGC TTC AG	Rat
*Myh7*	GGG TAT CCG CAT CTG TAG GA	TTG GTG TGG CCA AAC TTG TA	Rat
*Atgl*	TAC TGA AGA CCC TGC CTG CT	TGG CAA GTT GTC TGA AAT GC	Rat
*Cpt1*	AGG CCT CCA TGA CAA GAA TG	GTC ATG GCT AGG CGG TAC AT	Rat
*Acadvl*	TGA CCC TGC CAA GAA TGA CT	GTC ATG CAT GCC CAC AAT CT	Rat
*Acadm*	CCG GGA CTA GGG TTT AGC TT	ACT CTC CGG AAT GTG TGT GT	Rat

## Data Availability

The data presented in this study are available on request from the corresponding author.

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
