# Peer review of "Letrozole Accelerates Metabolic Remodeling through Activation of Glycolysis in Cardiomyocytes: A Role beyond Hormone Regulation"

_ijms, 2022, doi:10.3390/ijms23010547_

Round 1
Reviewer 1 Report
Interesting study. For table 1 and table 2 I would use a smaller font especially for the description of the primers. 208 and 308: I would put the sentences in the discussion. In the figures I would not leave a p <0.05 for everything, but I would show p = in order to give more meaning to one comparison rather than another.
Author Response
Comments and Suggestions for Authors
Interesting study.
For table 1 and table 2: I would use a smaller font especially for the description of the primers.
Following the reviewer’s suggestion, we arranged letters in tables to a smaller font.
208 and 308: I would put the sentences in the discussion.
We have put the sentences to summarize the results, and the sentences have already been reflected in discussion part, again.
In the figures: I would not leave a p <0.05 for everything, but I would show p = in order to give more meaning to one comparison rather than another.
We also agree with the reviewer’s opinion, because the suggested notation, such as “p =”, shows more apparent comparison between two. However, we consider that our manner, such as “p<0.05” could enough represent meaningful difference.

Reviewer 2 Report
This manuscript is an original research manuscript, which is devoted to the study of the effects of letrozole on the cardiovascular system. The manuscript is written very well. The research results have potentially high clinical significance.
The paper is suitable for publication. No further corrections are needed.
Author Response
Comments and Suggestions for Authors
This manuscript is an original research manuscript, which is devoted to the study of the effects of letrozole on the cardiovascular system. The manuscript is written very well. The research results have potentially high clinical significance.
The paper is suitable for publication. No further corrections are needed.
Thank you for your favorable consideration.
Reviewer 3 Report
Dear Editor,
The manuscript submitted by Heo and Co. is a study of the effect of estrogen blockers on cardiomyocyte metabolism. The study is well designed and conducted.
Some minimal issues need to be addressed before the manuscript is ready for publication in IJMS.
The first problem concerns the concept of the study. The authors need to add a little more information (for a wide range of IJMS readers) about the association between breast cancer, cancer chemo-radiotherapy and metabolic remodelling of the heart.
Is there any information about the combined effect of AI and chemotherapy/ AI and radiotherapy in breast cancer patients?
It would be great if the authors could add more information about cardiovascular disease in the introduction. Which type of cardiovascular diseases are described in patients with AI in references 8-10 (page 1, line 37)?
It would be important to explain how the optimal concentration of letrozole (0.1 mg/kg/day) was selected and why the treatment was given in a 28-day time window. Was a cell toxicity assay performed on cardiomyocytes with different AI concentrations? Is there any information in the literature about the concentration? In any case, the authors should mention this.
The authors need to propose a mechanistic link for their assessments... and add some sentences about the molecular pathway underlying the effect…..how can AI (estrogen/estrogen inhibitor) affect lipid metabolism? What is the proposed signalling pathway and how/why is it affected?
AI affects the downstream target genes and proteins of PPAR alpha, PPAR alpha is also regulated by phosphorylation..it is known that the increased phosphorylation of PPAR alpha has an inhibitory effect on PPAR alpha transcriptional activity...is there any information (including the results of the present study) about the changes in PPAR alpha (mRNA, Protein full length) and phosphorylation of PPAR alpha after AI treatments? Did the authors investigate this by immunoblotting?
Is there any information about the beneficial effects of PPAR alpha agonists in the context of AI?
The same is valid for AI and alteration in structural proteins Myh6, Myh7 and ATP2a2…what is the molecular mechanism of these alterations due to estrogen and estrogen inhibitors? (even hypothetical)
I hope, addressing these issues can improve the manuscript.
Sincerely
Author Response
Reviewer 3.
The manuscript submitted by Heo and Co. is a study of the effect of estrogen blockers on cardiomyocyte metabolism. The study is well designed and conducted.
Some minimal issues need to be addressed before the manuscript is ready for publication in IJMS.
We have made the following editorial changes in response to the reviewer’s comments.
The first problem concerns the concept of the study. The authors need to add a little more information (for a wide range of IJMS readers) about the association between breast cancer, cancer chemo-radiotherapy and metabolic remodeling of the heart.
Various chemotherapeutic drugs including cyclophosphamide and doxorubicin were used in breast cancer patients (PMID: 32550797). Furthermore, as adjuvant drugs, AIs, anastrozole and letrozole, and/or SERMs, tamoxifen, are commonly used in ER+ breast cancer which is one of the subtype of breast cancer, to block estrogenic influence in tumor cell’s proliferation (PMID: 21709140). And, it is revealed that these patients, suffering for ER+ breast cancer and taken to antiestrogen therapy, is more vulnerable to cardiovascular diseases (PMID: 27998966). The underlying cause has been attributed to changes in lipid metabolism in the heart in response to circulating estrogens levels (PMID: 29331535, 29331535). This perspective seems reasonable as estrogens play a role in energy and metabolic homeostasis (PMID: 21511884). However, it is not clear that chemotherapeutic drugs and radiotherapy have a cardiotoxicity between cardiac metabolism, although doxorubicin used as a chemotherapeutic drug is thought to have an effect on heart by reducing mitochondrial respiratory efficiency and cytosolic adenosine triphosphate content coupled with an increase in free radical formation hypothetically (PMID: 30444915, 21094500).
(Introduction)
As a first line of breast cancer therapy, chemotherapy drugs are toxins and can induce cardiac toxicity. As a results of chemotherapy, the heart function could reduce and limit enough oxygen and nutrients to supply the body. Along with chemotherapy drugs, anti-estrogenic therapy could be selected in ER+ breast cancer patients. Aromatase inhibitors are helpful for women who have gone through menopause because they stop most estrogen production in the body, including ovary and fat.
Is there any information about the combined effect of AI and chemotherapy / AI and radiotherapy in breast cancer patients?
Conventionally, AIs have been used as an adjuvant drugs. The first line therapeutic drugs are chemicals, such as cyclophosphamide and doxorubicin. The context would have been plentiful through corrected sentences to first question.
It would be great if the authors could add more information about cardiovascular disease in the introduction. Which type of cardiovascular diseases are described in patients with AI in references 8-10 (page 1, line 37)?
Following the reviewer's suggestion, we actually add the information required in the original text.
(Introduction)
“However, cohort studies have reported that patients taking AIs—rather than tamoxifen—are more predisposed to cardiovascular diseases including myocardial infarction and heart failure.”
It would be important to explain how the optimal concentration of letrozole (0.1 mg/kg/day) was selected and why the treatment was given in a 28-day time window. Was a cell toxicity assay performed on cardiomyocytes with different AI concentrations? Is there any information in the literature about the concentration? In any case, the authors should mention this.
Our present animal experimental design was impressed from previous researches which used letrozole-induced polycystic ovarian syndrome (PCOS) rats as an animal model (PMID: 30287740, 30524282). The studies showed enough induced PCOS state which represents low circulating estrogens level, when rats were exposed to dose on 0.1mg/kg/day for 28-days. Therefore, we mimicked the studies and also observed the PCOS state enough developed in our present study too (Figure 1). Furthermore, we also referred to several researches, which used letrozole in vitro experiment, to decide the proper concentration in our vitro assay (PMID: 32006244, 33182361). Subsequently, we performed toxicity test on H9c2 cardiomyocytes with some letrozole dose (0nM, 50nM, 100nM, 500nM, 1uM). After observation of the cell’s viability, we chose the 100nM of letrozole concentration, since the dose is selected generally and showed no observed adverse effect (NOAEL) in our experimental doses (Under the figure).
The authors need to propose a mechanistic link for their assessments... and add some sentences about the molecular pathway underlying the effect... how can AI (estrogen/estrogen inhibitor) affect lipid metabolism? What is the proposed signaling pathway and how/why is it affected?
We expected that the alteration in cardiac lipid metabolism is blamed to low estrogens via AI because E2-treated cells show identical changes in vivo. While the letrozole-induced rats showed increased TG accumulation and decreased oxidation and uptake of FA, letrozole fails to change the lipid metabolism without supplements of steroids. Since estrogens cooperated with estrogen receptor modulate transcription of PPARα, which is known to master factor regulating transcription related to lipid metabolism, the letrozole-induced rats in this study reflect estrogen inhibition. Therefore, the low estrogenic environment decreases FA oxidation and uptake by down-regulation of PPARα signaling.
Following the reviewer's suggestion, we corrected a sentence to include a more detailed description above explained context in the discussion.
(Discussion)
"These findings suggest that estrogens can modulate lipid metabolism by regulating estrogen receptor signaling and PPARα signaling, and thereby, might be induce cardiac structural remodeling from altered lipid metabolism.”
AI affects the downstream target genes and proteins of PPAR alpha, PPAR alpha is also regulated by phosphorylation. it is known that the increased phosphorylation of PPAR alpha has an inhibitory effect on PPAR alpha transcriptional activity...is there any information (including the results of the present study) about the changes in PPAR alpha (mRNA, Protein full length) and phosphorylation of PPAR alpha after AI treatments? Did the authors investigate this by immunoblotting?
Following the reviewer's suggestion, we confirmed the expression level of PPARα using a method of quantitative PCR (Under the figure). We observed lower PPARα expression levels in letrozole-induced rats than vehicle-treated rats. While the PPARα expression level was increased in estradiol (E2)-treated cells, the level was not altered in letrozole-treated cells. These results represent that estrogens cooperated with estrogen receptors act as a transcription factor of PPARα (PMID: 12914523).
Is there any information about the beneficial effects of PPAR alpha agonists in the context of AI?
Following the reviewer's suggestion, we investigated additional information about the PPARα agonist. Fibrates, one of the PPARα agonists, increase FA oxidation (PMID: 29662003). As one of fibrates members, fenofibrate inhibits aromatase expression (PMID: 12576508). Therefore, we consider fenofibrate a helpful drug to combine with AIs to protect against cardiovascular diseases. Moreover, we think there might be a possibility that fenofibrate has an effect as target drug of ER+ breast cancer, although it needs to be warranted through further study.
(Discussion)
Therefore, it also might be intimate that PPARα agonist including fenofibrate, as an activator of fatty acid oxidation, might have an effect as target drug of ER+ breast cancer, although it needs to be warranted through further study.
The same is valid for AI and alteration in structural proteins Myh6, Myh7 and ATP2a2…what is the molecular mechanism of these alterations due to estrogen and estrogen inhibitors? (even hypothetical)
We have considered that cardiac structural change is a compensatory reaction caused by insufficient energy. For more muscular contraction, cardiomyocytes change the structure of the cells and mediate the related factors, including Myh6, Myh7 and ATP2a2. However, the association between AIs and the factors is not established until now. In the present study, the possibility is suggested the increased glycolysis by the estrogen-independent role of letrozole.
I would like to thank you for your insightful and thoughtful comments.

Reviewer 4 Report
Manuscript ID: ijms-1459640
In this study by Jun Heo et al, the authors claim to show the hypertrophic changes in the left ventricle and the metabolic energy changes associated with decreased FA oxidation and increased glycolysis in the heart of female rats exposed to the AI, letrozole.
I have some comments that I would like to be addressed:
Regarding figure 2: the authors state an hypertrophic response. Please add cross section of the hearts that show an increase of wall-thickness. Also a quantitative measurement of cardiomyocyte size increasement is demanded when the authors decide to make this claim.
Regarding the results presented in figure 3. The authors state that their findings indicate increased TG accumulation and decreased TG degradation and FA oxidation, thereby suggesting that FA consumption by cardiomyocytes is lower in the presence of letrozole-induced reduced estrogens levels. But this is a very condensed summary of what is happening in the animals on one hand and the H9C2 cells on the other. For example, LTZ has a clear effect on Atg1, Cpt1, Acadvl and Acadm in the animals, whilst is absent in the cells. The effect of E2 is only shown in cells, that don’t reflect the reducing effect of LTZ. The same holds for CD36 expression. This is strange, please explain what is happening here and why these two systems differ from eachother. It is not very nice to make a statement only on the results that are pointing towards the hypothesis.
The title in line 273: 3.3. Letrozole induces enhanced glucose consumption in cardiomyocytes independent of its role in modulating the estrogens levels is to bold, since the authors do not directly measure differences in estrogen levels, only a surrogate of measuring mRNA expression changes.
Lines 285-289: This is very suggestive. Is the increase in E2 cells significantly lower than in LTZ cells? If not than this effect seems not to be induced by estrogen.
Regarding figure 4: The authors state that “letrozole enhances glycolysis in cardiomyocytes, independent of its role in modulating the estrogens levels, to compensate for decreased FA oxidation.” For me it is not clear how the authors demonstrate that this is independent of its role in modulating the estrogens levels. I see only 2 conditions changed: +/-LTZ and H/L glucose.
Line 400: The data does not show the estrogen blocking capacity of letrozole
Minor:
Line 399: induce promotion : remove one of the two
Author Response
Reviewer 4.
In this study by Jun Heo et al, the authors claim to show the hypertrophic changes in the left ventricle and the metabolic energy changes associated with decreased FA oxidation and increased glycolysis in the heart of female rats exposed to the AI, letrozole.
I have some comments that I would like to be addressed:
We have made the following editorial changes in response to the reviewer’s comments.
Regarding figure 2: the authors state a hypertrophic response. Please add cross section of the hearts that show an increase of wall-thickness. Also a quantitative measurement of cardiomyocyte size increasement is demanded when the authors decide to make this claim.
We also agree with the reviewer's opinion, which shows the cross-section of hearts to assess the wall thickness. The cross-section would be needed a sagittal plane to assess the wall-thickness quantitatively. However, unfortunately, we do not have the sagittal plane and just have a transversal plane of hearts which is inappropriate for assessing the wall-thickness and cardiomyocyte size.
Nevertheless, we observed that the ventricular wall, cytoplasm volume, and structural protein markers (Myh6, Myh7 and ATP2a2) were substantially increased in the letrozole-treated rats. Please understand our situation.
Regarding the results presented in figure 3: The authors state that their findings indicate increased TG accumulation and decreased TG degradation and FA oxidation, thereby suggesting that FA consumption by cardiomyocytes is lower in the presence of letrozole-induced reduced estrogens levels. But this is a very condensed summary of what is happening in the animals on one hand and the H9C2 cells on the other. For example, LTZ has a clear effect on Atg1, Cpt1, Acadvl and Acadm in the animals, whilst is absent in the cells. The effect of E2 is only shown in cells, that don’t reflect the reducing effect of LTZ. The same holds for CD36 expression. This is strange, please explain what is happening here and why these two systems differ from each other. It is not very nice to make a statement only on the results that are pointing towards the hypothesis.
As an aromatase inhibitor, letrozole is a drug that is a blocker to produce estrogen from androgens. Letrozole has been used as an anticancer therapeutical drug for estrogen receptor-positive (ER+) breast cancer patients. After administration from letrozole, the rats could be reflected in the states of hypoestrogenism and hyperandrogenism. We confirmed the reduced estrogenic environment, as shown in figure 1. The letrozole-treated rats also revealed high TG accumulation and low lipid oxidation in the heart. There are two possibilities of lipid metabolism after letrozole treatment; one is mediated with sex hormones (low estrogen level and high androgen level), and another is directly involved with letrozole itself. To separate the two effects of letrozole, we introduced in vitro system. To avoid the steroid contamination of FBS, we used the charcoal dextran treated FBS, and treated sex steroid hormones into H9C2 cells. Like animal models having hypoestrogenism and hyperandrogenism, estrogen treatment induced the Atg1, Cpt1, Acadvl and Acadm expressions. However, letrozole and androgen treatments were similar to the control group. We concluded that low TG degradation and lipid oxidation seem to be considered by the low plasma estrogen (figure 3E).
Figure 3.
The title in line 273: 3.3. Letrozole induces enhanced glucose consumption in cardiomyocytes independent of its role in modulating the estrogens levels is to bold, since the authors do not directly measure differences in estrogen levels, only a surrogate of measuring mRNA expression changes.
We agree with the reviewer’s opinion because estrogens level is an indicator to confirm whether letrozole has been sufficiently effective. Based on our previous studies (PMID: 30287740, 30524282), this animal model might have characteristics with hypoestrogenism and hyperandrogenism. We, therefore, confirmed the estrogenic environment in letrozole inserted rats in this study.
Lines 285-289: This is very suggestive. Is the increase in E2 cells significantly lower than in LTZ cells? If not than this effect seems not to be induced by estrogen.
It is interesting because alteration of lipid metabolism depends on reduced circulating estrogens levels by letrozole. However, we observed that glycolytic alteration was not originated by the circulating estrogens levels but the letrozole effects. Therefore, letrozole seems to have another role in cardiac energy metabolism, except estrogens level regulator.
Regarding figure 4: The authors state that “letrozole enhances glycolysis in cardiomyocytes, independent of its role in modulating the estrogens levels, to compensate for decreased FA oxidation.” For me it is not clear how the authors demonstrate that this is independent of its role in modulating the estrogens levels. I see only 2 conditions changed: +/-LTZ and H/L glucose.
We have considered the two roles of letrozole in cardiomyocytes increasing glycolysis. One is a compensatory reaction to estrogens-mediated decreased consumption of fatty acids. The imperfection of FA oxidation induces glucose consumption to maintain the physiological heart’s pumping function through many kinds of research (PMID: 10408759, 8067430, 28395011). The other is the estrogens independent reaction demonstrated by the 2 conditions: +/-LTZ and H/L glucose in the present study. Moreover, we agree with the reviewer's opinion, 2 condition is insufficient to demonstrate the AI’s independent effect of letrozole in cardiac glycolysis. So we also performed glucose stress test, which could quantitatively measure the consumed glucose capacity; ECAR, and observed the increased ECAR level. We think this increased ECAR level could demonstrate being mediated by estrogens independent itself reaction.
Line 400: The data does not show the estrogen blocking capacity of letrozole
We expect LTZ-induced rats to develop the polycystic ovarian syndrome, representing the states of hypoestrogenism and hyperandrogenism, when the drug enough exerts estrogen-blocking effect, through our previous study (PMID: 30287740, 30524282). Furthermore, we have considered that figure 1 describes the estrogen-blocking capacity of letrozole in LTZ-induced rats, although the direct estrogens level was not measured.
Minor:
Line 399: induce promotion: remove one of the two.
Following reviewer’s indication, we revised the sentence.
(Conclusion)
“Our data announce that letrozole is an independent molecule to promote glucose oxidation as well as an estrogens blocker in cardiomyocytes.”

Round 2
Reviewer 4 Report
I believe that this manuscript would benefit from the additional experiments as proposed in my first comments. Now no substantial solutions were found to take away my doubts.